# Ultrasensitive single-step CRISPR detection of monkeypox virus in minutes with a vest-pocket diagnostic device

Yunxiang Wang[1,4], Hong Chen[1,4], Kai Lin[2,4], Yongjun Han[1,4], Zhixia Gu[3], Hongjuan Wei[1], Kai Mu[1], Dongfeng Wang[1], Liyan Liu[1], Ronghua Jin ®[3] ✉, Rui Song[3] ✉, Zhen Rong ®[1] ✉ & Shengqi Wang ®[1] ✉

The emerging monkeypox virus (MPXV) has raised global health concern, thereby highlighting the need for rapid, sensitive, and easy-to-use diagnostics. Here, we develop a single-step CRISPR-based diagnostic platform, termed SCOPE (Streamlined CRISPR On Pod Evaluation platform), for field-deployable ultrasensitive detection of MPXV in resource-limited settings. The viral nucleic acids are rapidly released from the rash fluid swab, oral swab, saliva, and urine samples in 2 min via a streamlined viral lysis protocol, followed by a 10-min single-step recombinase polymerase amplification (RPA)-CRISPR/Cas13a reaction. A pod-shaped vest-pocket analysis device achieves the whole process for reaction execution, signal acquisition, and result interpretation. SCOPE can detect as low as 0.5 copies/μL (2.5 copies/reaction) of MPXV within 15 min from the sample input to the answer. We validate the developed assay on 102 clinical samples from male patients / volunteers, and the testing results are 100% concordant with the real-time PCR. SCOPE achieves a single-molecular level sensitivity in minutes with a simplified procedure performed on a miniaturized wireless device, which is expected to spur substantial progress to enable the practice application of CRISPR-based diagnostics techniques in a point-of-care setting.

Since the outbreak of monkeypox (Mpox) in 2022, 116 countries and regions have reported more than 90000 cases to the World Health Organization (WHO). In July 2022, Mpox was listed as a "public health emergency of international concern". Mpox is a viral zoonosis caused by monkeypox virus (MPXV) infection, which behaves similarly to smallpox and is accidentally transmitted to humans when encountering infected animals[1]. Symptoms of Mpox infection in humans include fever, headache, muscle soreness, back pain, swollen lymph nodes, and skin rash[2]. Most patients recover within a few weeks, but some patients become seriously ill or even die. So far, two major strains have been found in Central Africa and West Africa, and the former is related

to more serious diseases[1,3]. MPXV strains found in Portugal are related to virus strains found mainly in West Africa. Compared with the strains prevalent in Central Africa, this strain causes less disease and lower mortality[4] – about 1% of the poor rural population[5].

MPXV is a zoonotic DNA virus that was first described in humans in the Democratic Republic of the Congo in 1970[6]. MPXV spreads by being bitten by infected animals[7], large respiratory droplets, close or direct contact with skin lesions, and possibly through contaminated pollutants[8]. The current global outbreak of human MPXV infection shows that the biology of the virus, human behavior, or both have changed. These changes may be driven by the decline in smallpox

[1]Bioinformatics Center of AMMS, 100850 Beijing, China. [2]Department of Clinical Laboratory, Air Force Medical Center, Air Force Medical University, 100142 Beijing, China. [3]Beijing Ditan Hospital, Capital Medical University, 100015 Beijing, China. [4]These authors contributed equally: Yunxiang Wang, Hong Chen, Kai Lin, Yongjun Han. ✉e-mail: ronghuajin@ccmu.edu.cn; songruii@hotmail.com; rongzhen0525@sina.com; sqwang@bmi.ac.cn

immunity, the relaxation of COVID-19 prevention measures, and the recovery of international travel[8]. In addition to clinical data, the commonly used methods for detecting MPXV are laboratory tests, such as quantitative real-time polymerase chain reaction (qPCR)[9,10] and enzyme-linked immunosorbent assay[11]. However, qPCR requires expensive instruments and skilled technicians, which are not available in some areas with limited resources. Thus, the demand for efficient, convenient, and rapid detection methods in pathogen nucleic acid testing is rising rapidly.

Recently, clustered regularly interspaced short palindromic repeats (CRISPR)/Cas-based diagnostics (CRISPR-Dx) are perfectly compatible with point-of-care-testing (POCT) because of their rapid detection efficiency, excellent detection sensitivity, high specificity, and simple reaction conditions[12,13]. Nevertheless, the CRISPR/Cas system presents a limited sensitivity and efficiency without sample preamplification. Thus, the mainstream CRISPR-Dx assays obtain three cases: (i) The amplification-free CRISPR/Cas detection assay as an ideal nucleic acid detection tool, such as some electrochemistry[14–16], CRISPR-cascade[17,18], and digital microfluidic chip methods[19–21], exhibits competitive sensitivity. However, these techniques always rely on sophisticated devices and complex enzyme systems and are difficult to operate in resource-limited settings. (ii) A two-step CRISPR-Dx assay with target gene amplification and CRISPR trans-cleavage show PCR-comparable sensitivity. However, this two-step detection method is subjected to multistep transfer operation and has the risk of aerosol contamination, thereby limiting its popularity[22,23]. (iii) Due to the shortcomings of split-step methods, researchers develop a single-step CRISPR reaction[24,25], which synchronizes amplification and CRISPR reactions in a single-tube reaction. It can simplify the operation workflow, avoid amplicon contamination, and shorten the assay time, whereas generally sacrifice the reaction sensitivity to a certain extent. Given that CRISPR-based diagnostics are increasingly mature, making a quantum leap is difficult. Thus, researchers are focusing on ways to improve the performance of signal reporter units. Because of its low cost, easy operation, and excellent stability, lateral flow assay (LFA) exhibits as an ideal signal reporter for capturing the CRISPR cleavage products[26–28]. However, amplicon transfer may lead to the risk of aerosol contamination. Moreover, the CRISPR-LFA assay usually shows weakened sensitivity. Electrochemical sensors are highlighted as ultrasensitive signal transduction methods. Nevertheless, it is susceptible to environmental interference and is always costly. Fluorescence detection, as the most universal method, has been proven to have high sensitivity, fast speed, and excellent accuracy. However, most existing plate fluorescence readers are heavy, bulky, and expensive, thereby limiting their further application in resource-limited settings. Considering the need for point-of-care testing, the low-cost, rapid, and ultrasensitive signal transducer represents many untapped potentials.

To address the limitations of the current CRISPR-Dx, we develop a rapid, ultrasensitive, and field-deployable CRISPR/Cas13a diagnostic system, termed SCOPE (Streamlined CRISPR On Pod Evaluation platform), for point-of-care testing of MPXV in the rash fluid swab, oral swab, saliva, and urine samples within 15 min. We develop and systemically optimize a single-step recombinase polymerase amplification (RPA)-CRISPR/Cas13a assay, which could recognize the virus DNA as low as 0.5 copies/μL (2.5 copies/reaction). We intensively research the streamlined extraction-free viral lysis and reagent lyophilization to simplify the user operation and meet the on-site use requirements. A vest-pocket, multifunctional, and user-friendly homemade device called CRISPRPod (CPod) could accomplish the whole process without the aid of sophisticated instruments. We point a low-cost spectral sensor rather than an expensive optical filter in the CPod to collect the fluorescence signal. Finally, testing results of 102 clinical samples, including 35 MPXV samples, 19 herpes simplex virus (HSV) samples, and 48 negative samples, validate the clinical feasibility of SCOPE, demonstrating 100% sensitivity and specificity compared with qPCR.

SCOPE exhibits the potential to be utilized in resource-limited settings and could greatly enhance the MPXV diagnostic capabilities.

## Results

### Principle of SCOPE for MPXV detection

Herein, we reported the development of a rapid, ultrasensitive, and field-deployable CRISPR/Cas13a diagnostic system termed SCOPE for the point-of-care testing of MPXV in rash fluid swab, oral swab, saliva, and urine samples. As illustrated in Fig. 1a, the collected sample was treated with a 2-min extraction-free viral lysis, and followed by a 10-min optimized single-step RPA-CRISPR/Cas13a reaction, which incorporated RPA, T7 transcription, and Cas13a trans-cleavage in one tube. Through the unique trans-cleavage characteristic, the Cas13a protein was combined with crRNA to recognize the target nucleic acid sequence specifically and cleave RNA reporter group marked with quenched fluorescence nonspecifically to realize the accurate and efficient detection of the target. The whole process, including reaction incubation, fluorescent signal collection, and result interpretation, could be conducted with the aid of a homemade, vest-pocket, low-cost, and multifunctional device named CPod. The fluorescence signal could be detected by CPod via two different modes to satisfy different user demands. (i) Standalone mode: users could read the testing results by naked-eye just through the colors of indicator lights on the CPod. (ii) Wireless control mode: the real-time fluorescent curve and result interpretation of testing could be obtained by using a user-friendly smartphone application via wireless connection. As illustrated in Fig. 1b, the whole detection system just contained three Pasteur pipettes, a tube of lysis buffer, a tube of rehydration buffer, and a tube of single-step CRISPR lyophilized reagent. As shown in Supplementary Movie 1 and 2, users only need to load the collected sample into the lysis buffer and transfer the lysed sample with buffer into the single-step CRISPR reaction system. SCOPE could detect the MPXV within 15 min at resource-limited settings with a streamlined procedure and user-friendly workflow, which reduced the hands-on time and liquid-handling steps needed from less than 10 min and 5 pipetting steps in SHINEv.2 to less than 3 min and 3 user operations in this study[27].

### Development of SCOPE for MPXV detection

In this study, we established and systematically optimized a single-step RPA-CRISPR/Cas13a assay for the ultrasensitive detection of MPXV F3L gene DNA. Furthermore, a streamlined viral lysis and easy-operation reagent lyophilization with a user-friendly workflow was developed to meet the on-site or domestic use requirements.

### Single-step RPA-CRISPR/Cas13a assay

The single-step CRISPR assay incorporated RPA, T7 transcription, and CRISPR/Cas13a detection in one tube, which could avoid aerosol contamination and further simplify the workflow. As shown in Fig. 2a, the lysed MPXV DNA was amplified by RPA. The double-stranded DNA amplicons were labeled with T7 promoters at the 5′ end of forward primers, which enabled the transcription of single-stranded RNA in the presence of T7 RNA polymerase. The RNA products were recognized by the Cas13a and crRNA complex. Meanwhile, the trans-cleavage activity of Cas13a was activated and nonspecifically cut the fluorescence-quenching modified single-strand RNA reporter. The results could be interpreted by the fluorescence intensity.

For the single-step CRISPR reaction, the primer and crRNA screening significantly determines the detection sensitivity. First, we screened 16 groups of RPA primers, and the agarose gel electrophoresis indicated that the F2-R4 group exhibited the best amplification efficiency (Supplementary Fig. 1a). Then, 3 different crRNAs targeting MPXV F3L gene were designed and tested by the transcribed and purified F3L gene RNA. As illustrated in Supplementary Fig. 1b, crRNA-1 demonstrated the optimal cleavage activity. Nevertheless, when we

conducted the primers and crRNA screening simultaneously in single-step RPA-CRISPR/Cas13a reaction, it was found that the combination of best primers (F2-R4) and best crRNA (crRNA-1) didn't show the best detection efficiency. While, the combination of suboptimal primers (F3-R1) and suboptimal crRNA (crRNA-3) worked best (Supplementary Fig. 1c). We supposed it may stem from two aspects: (1) The single-strand DNA of primers and sing-strand RNA of crRNA may induce mutual interference; (2) The efficiency of amplification and Cas clea-vage should meet the balance. Thus, we recommended the screening of primers and crRNA in single-step CRIPSR should be conducted simultaneously to obtain the optimal combination.

Furthermore, to enhance the reaction efficiency and accelerate the reaction, several essential components of the assay system, including rNTP mix, T7 RNA polymerase, primers, LwaCas13a, crRNA, and RNase H concentration, were optimized iteratively with a single reagent modified in each experiment. The results indicated that 1 mM of rNTP mix, 1 U/uL of T7 RNA polymerase, 500 nM of primers, 50 nM

of LwaCas13a, 100 nM of crRNA, and 0.05 U/μL of RNase H generated the best performance (Fig. 2b–g).

## Extraction-free viral lysis

The extraction-free viral inactivation and lysis was systematically optimized to release the nucleic acids of the MPXV samples to sim-plify the workflow of sample pretreatment. According to the WHO, the best diagnostic specimens are swabs directly taken from the rash (skin, fluid, or crusts). Oropharyngeal, anal, or rectal swabs are alternative in the absence of skin lesions, and blood testing is not recommended (https://www.who.int/news-room/fact-sheets/detail/monkeypox). To further meet the on-site needs and simplify the sample collection, we selected rash fluid swab, oral swab, saliva and urine as the tested specimens in this study. And these four types of specimens collected from Mpox patients were utilized to optimize the lytic reaction. Temperature and time were the two critical para-meters of viral lysis, which significantly influenced the efficiency of

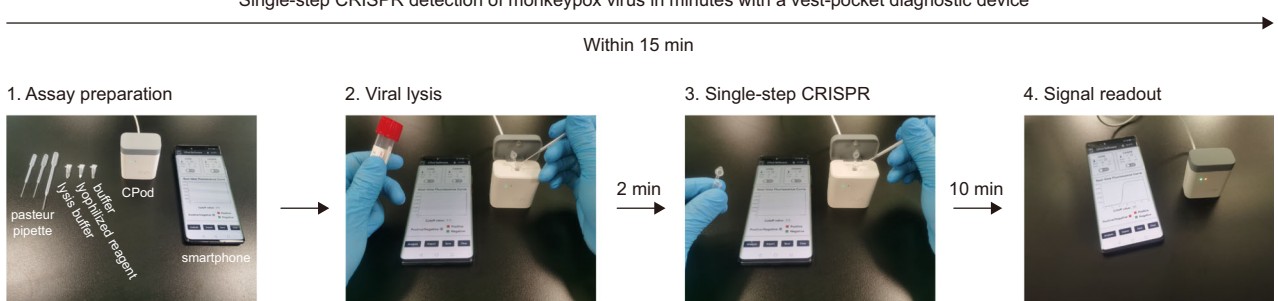

**Fig. 1 | Point-of-care testing of monkeypox virus using SCOPE. a** Schematic illustration of SCOPE for monkeypox virus detection. Rash fluid swab, oral swab, saliva, and urine samples were collected and processed with extraction-free viral lysis on the CPod at 80 °C for 2 min. The exposed monkeypox virus DNA was conducted with single-step RPA-CRISPR/Cas13a reaction at 37 °C for 10 min.

Qualitative interpretation could be made through the indicator lights of the CPod, or quantitative interpretation was carried out by a smartphone application via wireless connection. **b** The operation workflow of SCOPE for monkeypox virus detection.

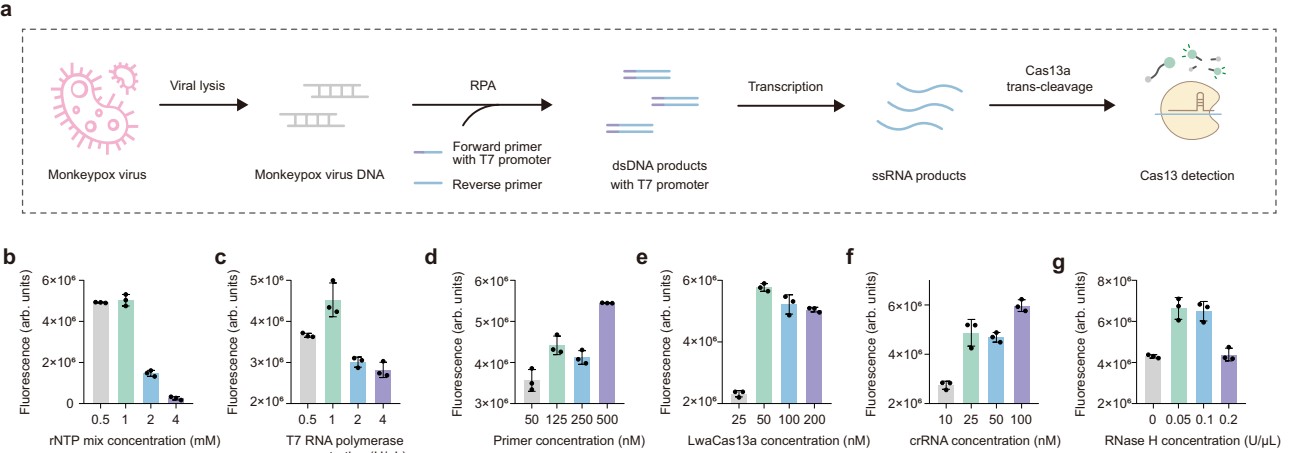

**Fig. 2 | Optimization of essential components of single-step RPA-CRISPR/Cas13a detection system. a** Schematic of the mechanism of single-step RPA-CRISPR/Cas13a assay. **b**–**g** The end-point fluorescence intensity of the single-step RPA-CRISPR/Cas13a detection for the optimization of six decisive reaction conditions, including the rNTP concentration of 0.5, 1, 2, and 4 mM; the T7 RNA polymerase concentration of 0.5, 1, 2, and 4 U/µL; primer concentration of 50, 125, 250 and 500 nM; LwaCas13a concentration of 25, 50, 100 and 200 nM; crRNA concentration of 10, 25, 50 and 100 nM; the RNase H concentration of 0, 0.05, 0.1 and 0.2 U/µL. Mean ± s.d. for 3 technical replicates for (**b**–**g**).

nucleic acid release. Figure 3a–d showed that several lysis temperatures, including room temperature, 40 °C, 80 °C, 95 °C, and 98 °C, are separately tested to obtain the optimum lysis temperature. All the data indicated that the reaction efficiency of these four types of specimens could attain saturation when the lysis temperature reached above 80 °C. Meanwhile, we optimized the viral lysis time at 80 °C, the lysis duration was optimized by setting 0, 2, 5, 10, and 15 min. As illustrated in Fig. 3e–h, the effect of 2-min lysis was equivalent to 5, 10, and 15 min. Therefore, incubation at 80 °C for 2 min was selected as the optimal lysis condition. Besides, we tested series of ratios of lysis buffer to clinical specimens, including 10:0, 9:1, 3:1, 1:1, 1:3, 1:9, 0:10. We determined the optimal ratio of lysis buffer to sample was 9:1 for rash fluid swab, 3:1 for oral swab, 1:3 for saliva, and 9:1 for urine (Fig. 3i–m). Furthermore, to evaluate the performance of our proposed assay with different lysed sample volume, we performed the single-step RPA-CRISPR/Cas13a reaction with 0, 1, 2, 5, and 7 µL of lysed samples. And all the four types of specimens exhibited the best reaction efficiency when using 7 µL of lysed samples (Fig. 3n–r). Compared with the multi-steps and time-consuming (>20 min) operation of nucleic acid extraction, our developed viral lysis only needed a 2-min incubation, demonstrating a superior potential for on-site use (Fig. 3s). To further test the performance of our lysis buffer, rash fluid swab, oral swab, saliva, and urine samples were processed using two methods—direct lysis and extraction with commercial kits. For single-step RPA-CRISPR/Cas13a detection of the treated virus, the results in Fig. 3t–v suggested that the direct lysis method obtained the similar result to the conventional nucleic acid extraction method for rash fluid swab, oral swab, and saliva specimens. However, due to the complex composition of urine sample, the lysis effect was remarkably lower than that of nucleic acid extraction (Fig. 3w).

## Reagent lyophilization

The lyophilization technology of the reaction system was explored to facilitate the transportation and storage of the reaction regents and reduce artificial experimental procedures (Supplementary Fig. 2a). The results in Supplementary Fig. 2b showed that untreated lyophilized reagent led to signal attenuation compared with the reagent before lyophilization. To blunt this attenuation due to reagent lyophilization, 0.1% bovine serum albumin (BSA) was added to the single-step CRISPR reagent as a cryoprotectant additive.

## Fabrication of the vest-pocket analysis device

A miniaturized, lost-cost, user-friendly, and multifunctional homemade device was designed and manufactured to adapt the above CRISPR-based detection assay (Fig. 4a–c). In order to minimize instrument size and save the material cost, the mechanical and electronical components (Supplementary Fig. 3–4) were under meticulous design. As illustrated in Fig. 4d, the metal ceramics heater and PT1000 temperature sensor were controlled by ESP32 microcontroller unit via proportion integration differentiation algorithm. Furthermore, a cooling fan was appended to accelerate the cooling rate from lysis temperature (80 °C) to CRISPR reaction temperature (37 °C). The entire temperature control unit demonstrated an excellent performance in heating and cooling rate and temperature retention (Supplementary Fig. 5). In this study, a low-cost, tiny, and sensitive spectral sensor AS7341 rather than the conventional optical design with expensive optical filters was utilized in CPod to acquire the fluorescence signal. AS7341 spectral sensor integrated eight visible light channels and one near infrared spectrum channel could satisfy the requirement of multiplex detection. Consideration of just one target to be tested, only the Channel F4 of the AS7341 (FAM channel: 500–520 nm) was utilized. Specifically, the FAM-labeled fluorescence reporter could be excited by a blue LED, then the emission light could be collected by the spectral sensor. The real-time fluorescence intensity was recorded, analyzed, and interpreted. And the results were calibrated by the software algorithm to remove baseline offset (Supplementary Fig. 6). For our user-friendly CPod, the final diagnostic interpretation could be presented in two different modes that can be selected to meet the needs of individuals. For standalone mode (easy-to-use mode), the results could be readout by naked-eyes through the indicator lights. For wireless control mode (professional mode), CPod was wirelessly connected with a smartphone, and the collected real-time fluorescence intensity could be presented in an in-house designed smartphone application (Fig. 4e and Supplementary Fig. 7). The entire CPod was powered by a USB Type-C power adapter.

The proposed filter-free optical design can significantly reduce device size and instrument cost due to the utilization of the spectral sensor. The overall dimension of the homemade CPod was 52 mm × 30 mm × 60 mm. The total cost of the optical, mechanical, and electronic components was 13.46 dollars (Supplementary Table 1). This device can perform the entire workflow including reaction execution, signal acquisition, and result interpretation in either standalone mode or wireless control mode via smartphone application, which can

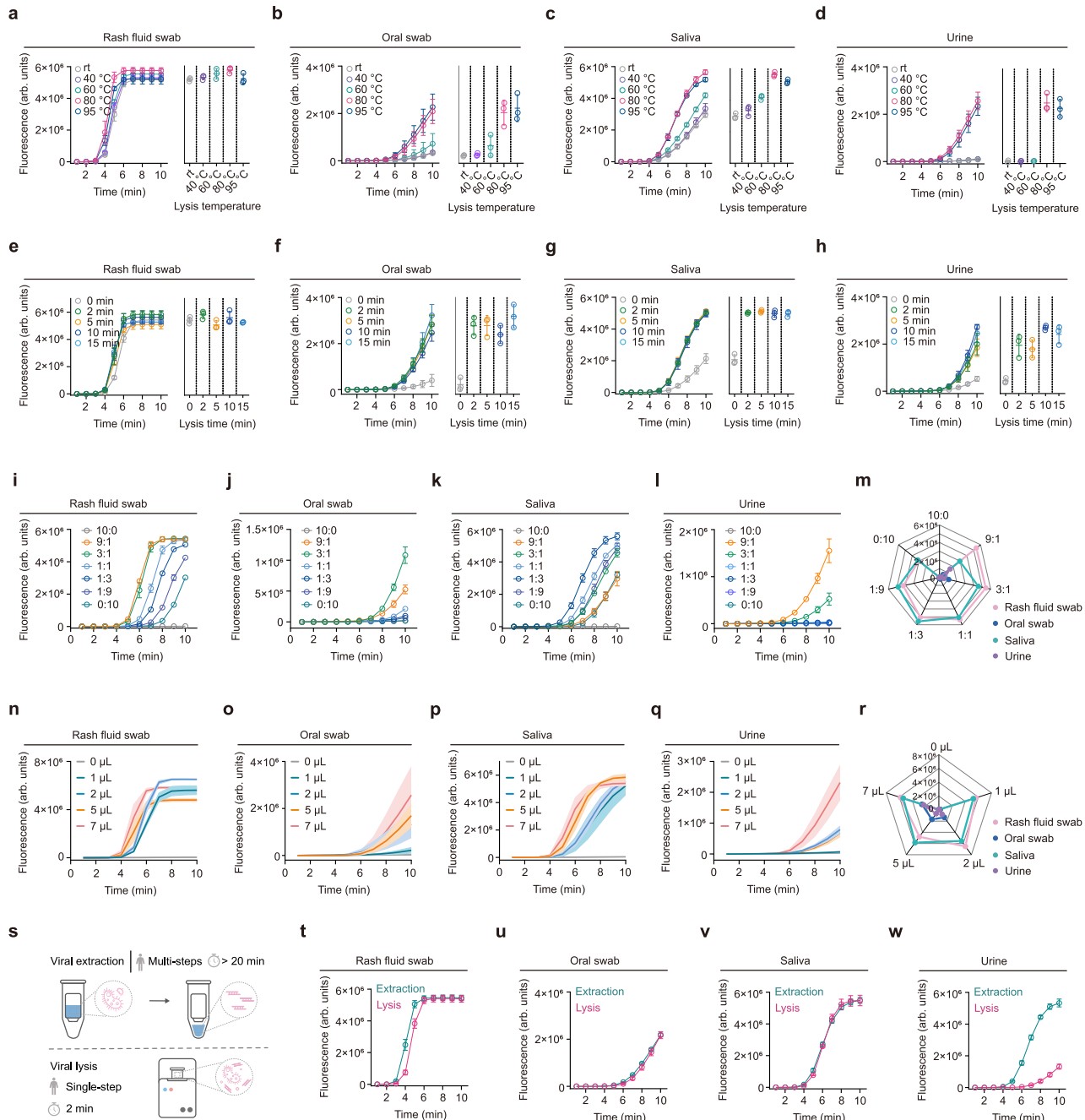

**Fig. 3 | Optimization of extraction-free viral lysis.** The real-time fluorescence curves and their end-point fluorescence intensity of the 10-minutes single-step CRISPR reaction using the lysed (**a**) rash fluid swab, (**b**) oral swab, (**c**) saliva and (**d**) urine samples with different lysis temperature (room temperature, 40, 80, 95, and 98 °C). Lysis time: 5 minutes. The real-time fluorescence curves and their end-point fluorescence intensity of the 10-minutes single-step CRISPR reaction using the lysed (**e**) rash fluid swab, (**f**) oral swab, (**g**) saliva and (**h**) urine samples with different lysis time (0, 2, 5, 10, and 15 minutes). Lysis temperature: 80 °C. The real-time fluorescence curves of the 10-minutes single-step CRISPR reaction using the lysed (**i**) rash fluid swab, (**j**) oral swab, (**k**) saliva and (**l**) urine samples and their (**m**) end-point fluorescence intensity with different ratios of lysis buffer to sample (10:0, 9:1,

3:1, 1:1, 1:3, 1:9, 0:10). The real-time fluorescence curves of the 10-minutes single-step CRISPR reaction loaded with different volumes of lysed (**n**) rash fluid swab, (**o**) oral swab, (**p**) saliva and (**q**) urine sample (0, 1, 2, 5, 7 μL), and their (**r**) end-point fluorescence intensity. (**s**) meant that the viral lysis was easy-to-use and time-saving than the conventional nucleic acid extraction. The real-time fluorescence curves of the 10-minutes single-step CRISPR reaction using (**t**) rash fluid swab, (**u**) oral swab, (**v**) saliva and (**w**) urine samples processed by viral lysis and nucleic acid extraction, respectively. For (**a-l**) and (**t-w**), error bars represent mean ± s.d. for 3 technical replicates. For (**n-q**), error bands represent mean ± s.d. for 3 technical replicates. In (**m, r**), values represent the mean for 3 technical replicates.

achieve superior analytical capabilities as standard desktop instruments. Since it doesn't require any exterior equipment, this device is regarded as a flexible and universal solution to meet the increasing molecular diagnostics needs at point of care. We believe this miniaturized, simple-to-use, and low-cost device possessed great application prospects in primary medical facilities.

## Analytical performance of SCOPE for MPXV detection

The limit of detection (LoD) of viral nucleic acids is crucial in early screening when the MPXV contained in the rash fluid swab, oral swab, saliva, or urine is at a low level. We determined the LoD of SCOPE with a dilution series of MPXV pseudovirus reference standard quantified by digital PCR. In Fig. 5a–c, the single-step RPA-CRISPR/Cas13a assay

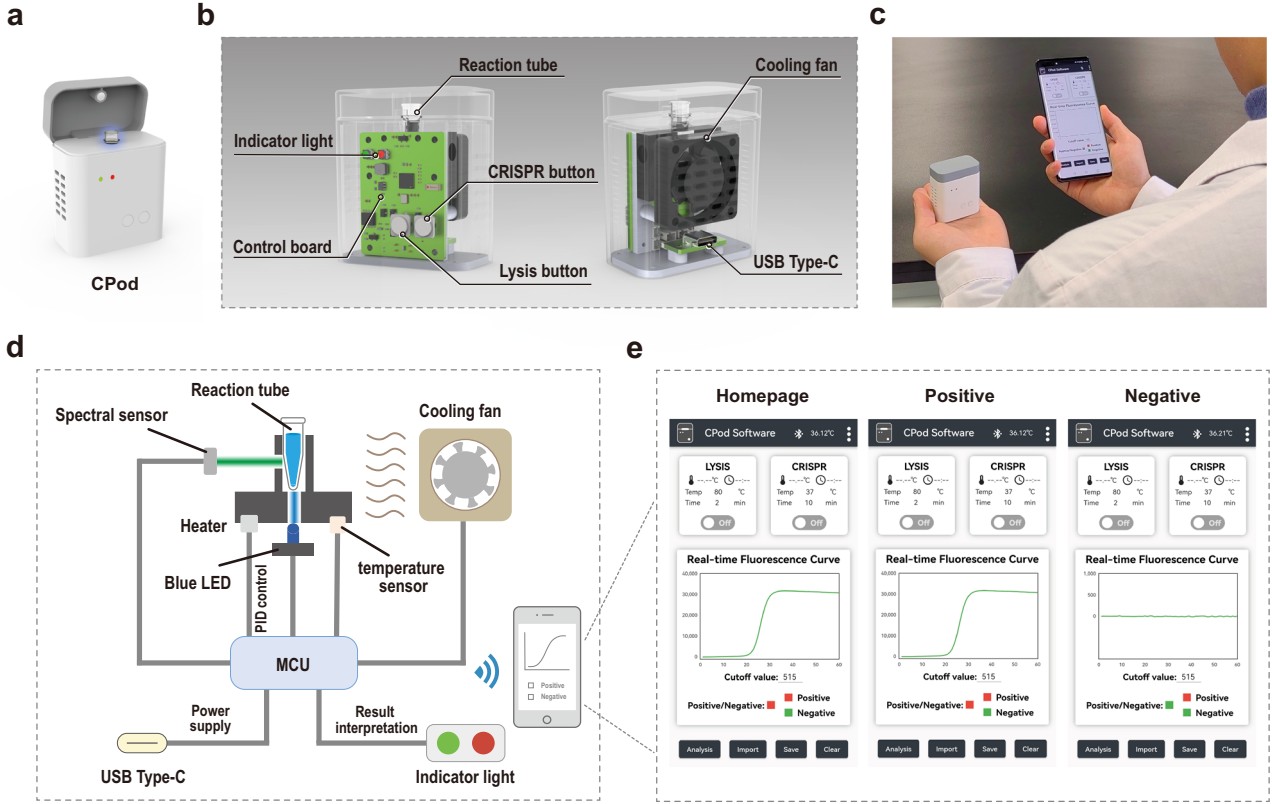

**Fig. 4 | Fabrication of the vest-pocket analysis device. a** The external rendering of CPod. **b** The interior rendering of CPod. **c** Photograph of vest-pocket CPod and wirelessly connected smartphone. **d** The schematic of the CPod working principle. **e** The smartphone application for the wireless control of CPod.

method was tested on the CPod and fluorescence plate reader respectively to detect a gradient of the corresponding MPXV DNA target at concentrations of 1000, 100, 10, 5, 1, 0.5, 0.2, and 0 copies/µL. The experimental results of homemade device and commercial instrument were recorded, and the final fluorescence values of each concentration detected by the two instruments were compared. The LoD tested by CPod was 0.5 copies/µL (2.5 copies/reaction), which was consistent with the LoD tested by the plate reader. Our tiny homemade instrument highlighted the detection capacity in the detection of low-concentration viral nucleic acid. The consequences demonstrated that our low-cost and compact equipment can achieve superior analytical capabilities as standard desktop instrument. This simple and effective detection platform was expected to facilitate its application in resource-limited settings. Besides, we performed a nonlinear regression fitting for the sensitivity testing results of CPod (Supplementary Fig. 8). Thus, we could estimate the input target concentration based on the fitting equation. To further explore and verify the sensitivity, reproducibility, and stability of SCOPE, 20 experiments were repeated independently with MPXV DNA at concentrations of 1, 0.5, 0.4, 0.3, 0.2, and 0 copies/µL. The threshold value was set based on the mean of the end-points fluorescence results of 20 NTC plus three times the standard deviation as 515. According to the threshold, the detection rates of positive samples were 100%, 100%, 80%, 45%, 15%, and 0%. We determined the positive detection rate greater than 95% as detectable concentration. Thus, the LoD was determined as 0.5 copies/µL (Fig. 5d–e and Supplementary Fig. 9). Meanwhile, several orthopoxviruses and other viruses with similar symptoms, including cowpox, vaccinia, and variola, herpes simplex virus (HSV), and varicella were validated without cross-reactivity with MPXV detection (Fig. 5f), thereby indicating the excellent specificity of our detection system.

## Validation by MPXV clinical samples

To investigate the clinical applicability of SCOPE, we first tested 24 mock saliva samples with the cycle threshold ($C_t$) value ranging from 27.86 to 41.86 (Supplementary Fig. 10a). All samples were correctly identified (Supplementary Fig. 10b) with a 100% concordance with qPCR assay (Supplementary Fig. 10c). To validate the robust clinical feasibility, a total of 102 clinical samples, including 35 Mpox positive samples (9 rash fluid swab samples, 9 oral swab samples, 9 saliva samples, and 8 urine samples), 19 HSV samples (5 rash fluid swab samples, 5 oral swab samples, 4 saliva samples, and 5 urine samples), and 48 negative samples from healthy males (12 rash fluid swab samples, 12 oral swab samples, 12 saliva samples, and 12 urine samples) were collected and tested by SCOPE side by side with the qPCR assay (Supplementary Fig. 11–13). As shown in Fig. 6a, qPCR with viral extraction took almost 2 h in total whereas our CRISPR-based assay with viral lysis only took 15 min. SCOPE correctly identified all 102 clinical samples with 100% sensitivity and 100% specificity (Fig. 6b, c). The $C_t$ values of the MPXV positive samples determined by qPCR assay ranged from 15.88 to 39.02. The results of SCOPE was 100% concordant with qPCR for all clinical samples (Fig. 6d, e), indicating that the proposed SCOPE for MPXV detection had competitive detection performance compared with the gold-standard qPCR assay, but with faster reaction speed, less user operation, and stronger field-deployable ability.

## Discussion

Recent outbreaks of infectious disease epidemics such as COVID-19 and Mpox have highlighted the necessity and importance of rapid, sensitive, and accurate nucleic acid POCT techniques for at-home self-test or in-field epidemic surveillance applications. Emerging CRISPR-Dx assays hold great potential to fulfill this pursue. However, most

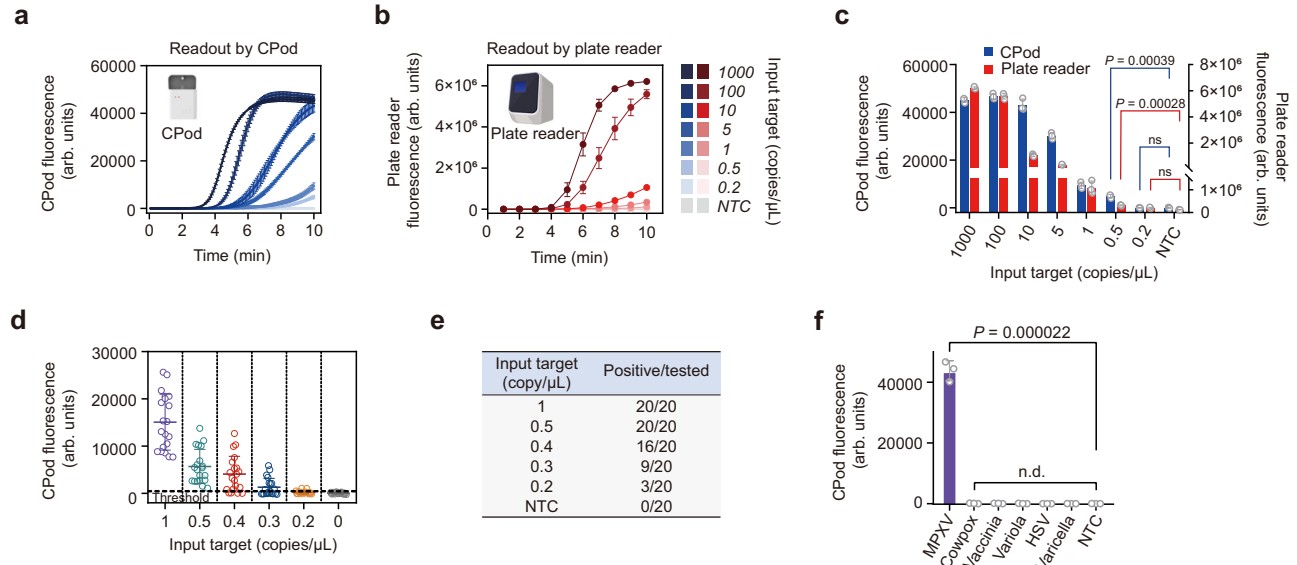

**Fig. 5 | Evaluation of the analytical performance of SCOPE.** Real-time fluorescence curves were tested by (**a**) CPod and (**b**) plate reader respectively and (**c**) their end-point fluorescence intensity for detection of monkeypox virus were collected with the DNA target at the concentrations of 1000, 100, 10, 5, 1, 0.5, 0.2, and 0 copies/μL. Sample volume: 5 μL. *** means $P < 0.001$. 'ns' means no significance, $P > 0.05$. **d** Reproducibility of our CRISPR-based system for the detection of monkeypox virus DNA at the concentrations of 1, 0.5, 0.4, 0.3, 0.2, and 0 copies/μL with independently repeating for 20 times. Sample volume: 5 μL. **e** Table of positive rates of testing with different concentrations of monkeypox virus DNA in (**d**). **f** Validation of the cross-reactivity between monkeypox virus (MPXV) detection and other orthopox virus and other viruses with similar symptoms detection including cowpox, vaccinia, variola, herpes simplex virus (HSV), and varicella using SCOPE. Templates concentration: $10^6$ copies/μL, sample volume: 5 μL. **** means $P < 0.0001$. 'n.d.' means not detected. For (**a**–**c**) and (**f**), mean ± s.d. for 3 technical replicates. For (**d**), mean ± s.d. for 20 technical replicates. For (**c**, **f**), $P$ values were calculated by two-tailed Student's t-test.

existing CRISPR-Dx assays use a two-step assay with nucleic acid pre-amplification and CRISPR trans-cleavage, which could lead to a complicated workflow and a risk of aerosol contamination. While as for current single-step CRISPR assays, there still existed a compromise between the analysis sensitivity and assay time. In this work, we developed a low-cost CRISPR-based POCT platform for MPXV detection with a qPCR-comparable detection performance but a faster speed, simplified workflow, and superior field-deployable ability. These superior features might address the bottleneck issues that limit the practice POCT application of CRISPR-Dx techniques.

Currently, most CRISPR-Dx methods utilized Cas12a for DNA target detection because Cas12a could directly target DNA amplicon without an additional transcription step. Nevertheless, as for single-step RPA-CRISPR reaction, the trans-cleavage activity of Cas12a with the assistance of crRNA could be activated once in the presence of DNA target, whereas the cis-cleavage activity will be also activated, leading to the unwanted cleavage of DNA target. In the early stage of DNA amplification, the loss of target templates will hinder the nucleic acid amplification and finally result in poor detection efficiency (Supplementary Fig. 14a). As a contrast, Cas13a has been proved with stronger collateral activity[29]. Besides, in the single-step CRISPR reaction, Cas13a and crRNA complex targeted the transcribed RNA, therefore the activated cis-cleavage activity just cuts the RNA target, which will not affect DNA templates or amplicons (Supplementary Fig. 14b). Furthermore, physical or chemical isolation of nucleic acid amplification from Cas detection has been validated to significantly improve the reaction sensitivity[30,31]. The additional T7 transcription step could delay the activation of Cas13a to weaken the interference between RPA and Cas13a cleavage. Thus, we believed Cas13a was preferable to Cas12a in single-step CRISPR reaction for DNA target detection.

Herein, we successfully developed a single-step RPA-CRISPR/Cas13a assay to detect MPXV at the single-molecule level in 10 min by systematically optimizing the assay conditions to reduce the interference between RPA and CRISPR reactions. As shown in Supplementary Table 2, compared with the existing CRISPR-Dx assays, SCOPE possessed the shortest reaction time from sample input to answer out and competitive sensitivity[16,17,19,22–25,27,32–46]. As shown in Supplementary Table 3, compared with the existing MPXV detection assays (excluding PCR-based methods)[47–56], SCOPE showed the lowest LoD and shortest turn-around time. Besides, the utilization of viral lysis and reagent lyophilization made it easy-to-use, time-saving, and suitable for on-site use. Furthermore, most of the existing studies were lacking of clinical verification, whereas we tested 102 clinical samples (35 Mpox cases, 19 HSV cases, 48 negative cases), showing superior clinical feasibility. Due to its excellent detection performance, we anticipate SCOPE will serve as the iconic detection sensitivity and turn-around time of CRISPR-Dx techniques.

In addition, specialized nucleic acid extraction steps, cold-chain logistics for test kit distribution and storage, and bench-top laboratory equipment are generally inaccessible to the end-users at resource-limit settings. Herein, a streamlined viral lysis method enabled the rapid and efficient nucleic acid release from rash fluid swab, oral swab, saliva and urine samples in 2 min. The test reagent mixture can be lyophilized without obvious decline in assay performance. Thus, the user's operation workflow can be simplified to meet the on-site use requirements. We further developed a vest-pocket wireless device CPod that could execute all detection procedures without the use of any additional instruments. Its filter-free optical design can significantly reduce instrument cost by employing a spectral sensor as the signal acquisition module. Notably, this design can be further extended for multiplex fluorescent analysis. This system could detect as low as 0.5 copies/μL (2.5 copies/reaction) of MPXV within 15 min from the sample input to the answer. The analysis data can be real-time readout and sent by the decentralized wireless CPod to the smartphone terminal for result interpretation and reporting.

Several further improvements can make our work more applicable at point-of-care needs. In the current workflow, a heating process at 80 °C for 2 min was required to release viral nucleic acid. The lysis buffer and reaction time should be optimized to achieve efficient virus

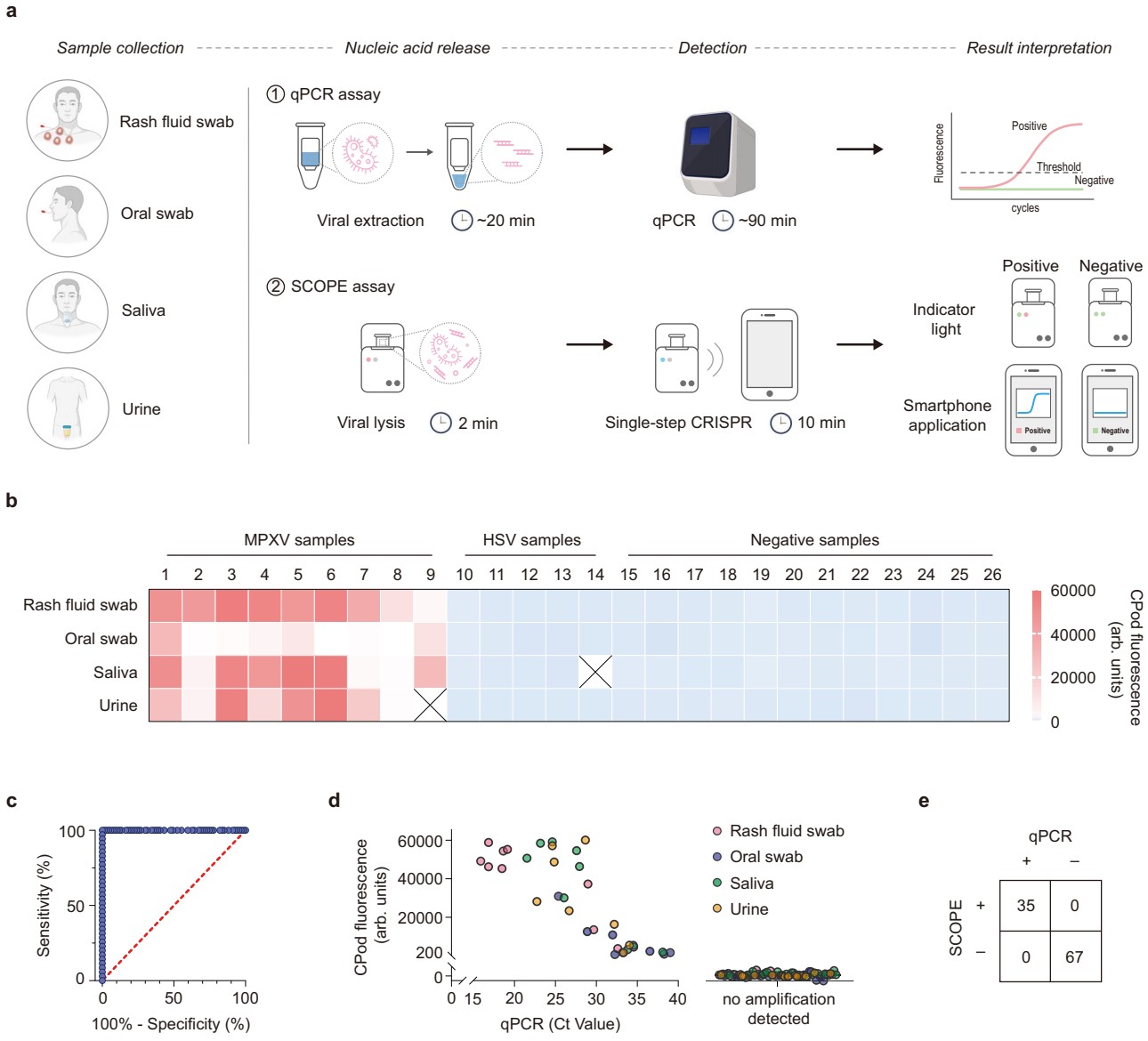

**Fig. 6 | Validation of the clinical feasibility of SCOPE. a** Workflow of qPCR assay and our CRISPR-based assay for testing of clinical samples. **b** The corresponding fluorescent intensities at 10 min for 102 clinical samples tested by SCOPE. **c** The receiver operating characteristic curve for 102 clinical samples tested by SCOPE.

**d** Clinical samples RT-qPCR $C_t$ value plotted against fluorescent intensity readout by CPod. **e** Correspondence of qPCR and SCOPE results for the monkeypox virus positive samples, herpes simplex virus (HSV) samples, and negative samples.

lysis at room temperature. Thus, the lysis process can be completed in a dropper for sample transfer similar to the way in lateral flow assay. Furthermore, a "sample-in, answer out" microfluidic chip can be designed to fully integrate sample lysis, solution transfer, and RPA-CRISPR reaction. Besides, the assay has only been validated using samples collected from male patients, and as such it is possible that the results cannot be extrapolated to non-male patients. We believe that SCOPE will be a powerful tool for point-of-care detection of MPXV and some other pathogens. This rapid and ultrasensitive single-step RPA-CRISPR assay can be readily incorporated with current Cas13a-based techniques to promote a substantial progress in CRISPR-Dx area.

## Methods
### Ethical statement
Clinical samples for this study were collected from the Beijing Ditan Hospital, and negative samples were collected from healthy volunteers, with informed written consent from all participants. This study was

performed following the guidelines approved by the Ethics Committee of Beijing Ditan Hospital (approval ID: DTEC-YJ2023-001-02).

### Clinical samples
All the Mpox clinical samples in this study were collected from male patients. According to epidemiological investigation, the 2022 Mpox epidemic was characterized by human-to-human spread outside of endemic areas, mostly in men who have sex with men[57], thus the herpes simplex virus samples and negative samples in this study were also collected from male patients as the negative control. The detailed information of clinical samples is listed in Supplementary Table 4. In detail, rash fluid swab and oral swab were collected and stored via disposable virus sampling tube. The saliva and urine were collected and stored via disposable sterile sampling cup.

### Materials
The materials used in this study were listed in Supplementary Table 5.

## Sample information

The MPXV, cowpox, vaccinia, and variola F3L gene pseudoviruses used in this study were synthesized by Fubio Biological Technology, Co., Ltd. (Suzhou, China) with quantitative analysis by digital PCR. The sequences of the above four F3L gene pseudoviruses were listed in Supplementary Table 7. The MPXV pseudovirus standard reference used for sensitivity and reproducibility evaluation was purchased from National Institute of Metrology, China.

## Mock clinical sample

The mock clinical samples in this study were adopted to evaluate the effect of viral lysis and clinical feasibility of our assay. Saliva samples were collected using the saliva collection tube from the healthy volunteers and spiked with the synthesized pseudovirus at different concentrations in a ratio of 9:1. Informed written consents from all participants were obtained.

## Viral lysis

The CRISPR-compatible viral lysis assay was developed and optimized to streamline operations. The collected clinical samples (7 μL) were mixed with chelex-100-based lysis buffer (20% chelex-100, 50 mM TCEP in TE buffer (10 mM Tris, 1 mM EDTA (pH 8.0))) in a ratio of 1:9 for rash fluid swab, 1:3 for oral swab, 3:1 for saliva, and 1:9 for urine. Then, the master mix was transferred to heating equipment or CPod followed by heating at 80 °C for 2 min.

## Viral DNA extraction

Clinical samples (1 mL) were firstly centrifuged at $14,000 \times g$ for 5 min, then gently removed 800 μL of supernatant. The remaining deposit was extracted by a commercial extraction kit. The released DNA was finally eluted by 200 μL of DNase/RNase-free water.

## Design of primers and crRNA

According to the previous study instruction, four pairs of RPA primers and three crRNAs were designed (Supplementary Table 6) to obtain a high reaction efficiency by Primer Premier 5 with a homologous sequence alignment via NCBI BLAST.

## crRNA preparation by in vitro transcription

For saving cost and time, the crRNAs used in this study were prepared by in vitro transcription (IVT). The ssDNA template with the reverse complementary sequence of the crRNA was synthesized with a T7 promoter (Sangon, China). An annealing reaction mix consisted of 1 μL of DNA template (100 μM), 1 μL of Standard Taq buffer (10×), 1 μL of T7-3G-IVT primer (100 μM, Supplementary Table 6), and 7 μL of DNase/RNase-free water. The master mix was denatured at 95 °C for 5 min, followed by gradient annealing with the ramp rate at 0.1 °C/s until cooling to 4 °C using a thermal cycler (Thermo, USA). Then, the annealing products were transcribed at 37 °C for 8 - 12 h via HiScribe T7 High Yield RNA Synthesis Kit according to the manufacturer's instructions. The transcribed crRNA with remaining DNA was treated with DNase I, followed by RNA purification via RNA Clean & Concentrator-5 Kit. Finally, the crRNA concentration was quantified by the Qubit™ Flex Fluorometer (Thermo, USA) via the Qubit™ RNA Broad Range Assay Kits, and stored at −80 °C until use.

## RPA primers screening

To evaluate the amplification efficiency of 16 pairs of primers, we performed RPA to amplify the MPXV DNA, followed by the agarose gel electrophoresis. For the RPA reaction, 50 μL master mix was prepared with 29.5 μL rehydration buffer, 500 nM forward and reverse primers, and 10 μL MPXV F3L gene DNA, then added with DNase/RNase-free water to 47.5 μL. Magnesium acetate (2.5 μL, 280 mM) was finally added to initiate the RPA reaction at 37 °C for 10 min. Then, the DNA amplicons of RPA were diluted 10 folds with DNase/RNase-free water and mixed with 6× DNA loading dye to 3% agarose gel electrophoresis. The gel image was captured by gel imaging system (Tanon 4600, China).

## crRNA screening

The Cas13a reaction for the crRNA screening was performed using the following condition: A mixture of 2 μL Tolo buffer 2 (10×), 50 nM Lwa-Cas13a, 100 nM crRNA, 1 μM ssRNA reporter, and 10 μL RNA target was added with DNase/RNase-free water to 20 μL. The RNA target of MPXV was produced by RPA-T7 transcription reaction, which was mixed with 1× RPA rehydration buffer, 500 nM forward and reverse primers, 1 U/μL T7 RNA polymerase, 1 mM rNTP mix, 0.05 U/μL RNase H, 1 U/μL RNase inhibitor. The prepared master mix was added to resuspend the RPA lyophilized pellet and divided into five 14-μL reaction units. Then, 5 μL of MPXV F3L gene DNA and 1 μL of magnesium acetate (280 mM) were added to activate the reaction. The RNA products were purified via RNA Clean & Concentrator-5 Kit.

## Detection of MPXV via qPCR assay

A qPCR assay for the detection of MPXV was performed using the following condition: 4 μL of Taq buffer (5×), 0.8 μL of Taq Master PCR Mix, 0.8 μL of forward primer (10 μM): MPXV-PCR-F (Supplementary Table 6), 0.8 μL of reverse primer (10 μM): MPXV-PCR-R (Supplementary Table 6), 0.8 μL of Taqman probe (10 μM): MPXV-PCR-P (Supplementary Table 6), and added DNase/RNase-free water to 20 μL. qPCR cycling conditions were as follows: hold at 50 °C for 2 min, polymerase activation at 95 °C for 2.5 min followed by 45 cycles with the denaturing step at 94 °C for 30 seconds, and annealing and elongation steps at 55 °C for 30 s. The real-time fluorescence intensity was monitored and analyzed by ABI-7500 (Thermo, USA).

## Bulk single-step RPA-CRISPR/Cas13a assay

70 μL of master mix was prepared with 1× RPA rehydration buffer, 50 nM LwaCas13a, 1 U/μL T7 RNA polymerase, 0.05 U/μL RNase H, 1 U/μL RNase inhibitor, 1 μM FAM-UUUUU-BHQ1, 1 mM rNTP mix, 500 nM forward primer: MPXV-RPA-F3 (Supplementary Table 6), 500 nM reverse primer: MPXV-RPA-R1 (Supplementary Table 6), and 100 nM transcriptional crRNA: MPXV-crRNA3 (Supplementary Table 6). The prepared master mix was added to resuspend the RPA lyophilized pellet and divided into five 14-μL reaction units. Then, 5 μL of DNA target and 1 μL of magnesium acetate (280 mM) were added to activate the reaction. Finally, the 20-μL master mix was transferred to the fluorescence plate reader (ABI-7500) or our developed CPod to acquire a 37 °C reaction condition and monitor the fluorescence intensity in real-time.

## Lyophilized single-step RPA-CRISPR/Cas13a assay

The lyophilized reagent was beneficial for transportation and storage, and facilitate point-of-care use. 50 μL of the master mix was prepared with 100 nM LwaCas13a, 2 U/μL T7 RNA polymerase, 0.1 U/μL RNase H, 2 U/μL RNase inhibitor, 2 μM FAM-UUUUU-BHQ1, 2 mM rNTP mix, 1 μM forward primer: MPXV-RPA-F3, 1 μM reverse primer: MPXV-RPA-R1, 200 nM crRNA: MPXV-crRNA3, and 0.1 % BSA. RPA lyophilized pellet was resuspended using the prepared master mix. Then, the resuspended mix was divided into five 10-μL reaction units followed by a snap-frozen at −50–−70 °C for 1 - 2 h and then was subjected to the vacuum system for 8 - 10 h (Labconco). The lyophilized reagent was stored at −20 °C until use. When in use, 15 μL of rehydration buffer (11.8 μL of RPA rehydration buffer, 2.2 μL of water, and 1 μL of 280 mM magnesium acetate) and 5 μL of the target were added to the lyophilized reagent. Finally, the 20-μL master mix was transferred to the fluorescence plate reader (ABI-7500) or our developed CPod to acquire a 37 °C reaction condition and monitor the fluorescence intensity in real-time.

## Clinical samples analysis

The rash fluid swab and oral swab were collected and stored in disposable virus sampling tube. The saliva and urine samples were

**Article** https://doi.org/10.1038/s41467-024-47518-8

collected and stored in disposable sterile sampling cup. The 7 µL of collected sample was added to lysis buffer in a 200-µL tube via Pasteur pipette at the ratio of 1:9 for rash fluid swab, 1:3 for oral swab, 3:1 for saliva, and 1:9 for urine. The lysis reaction was performed at 80 °C for 2 min on CPod. Then, 13 µL buffer and 7 µL lysed viral DNA were added to the lyophilized single-step CRISPR reagent in a 200-µL tube via Pasteur pipette and mixed well. The single-step CRISPR reaction was performed at 37 °C for 10 min on CPod. The results could be automatically interpreted and readout by indicator light or smartphone application of CPod.

## Data analysis

The schematics shown in Fig. 1a, Fig. 6a, and Supplementary Fig. 2a were created with BioRender.com. The data generated by the fluorescence plate reader (Applied Biosystem 7500 Real-Time PCR system) and the homemade device was exported and the data graphs were created using Graphpad prism 8 and Origin 2021. The gel image in Supplementary Fig. 1a was generated by Tanon 4600 gel imaging system. All the figures were grouped by Adobe Illustrator 2022. The homemade device was modeled by SOLIDWORKS 2017 and rendered by Keyshot 9. The circuits of the device were developed by using Cadence 22.1. The smartphone software was designed by using Flutter 3.16.

## Reporting summary

Further information on research design is available in the Nature Portfolio Reporting Summary linked to this article.

## Data availability

All the original data generated in this study are provided as Source Data. Source data are provided with this paper.

## Code availability

The code of the Android smartphone application is available at https://github.com/zayden/cpod. Bioinformatics Center of AMMS, Beijing, China owns the copyright of the code. The application is only for educational and research use. It is prohibited to use the code for any commercial purpose, including distribution, sale, lease, license, or other transfer to a third party.

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

## Acknowledgements

This work was supported by the Bioinformatics Center of AMMS.

## Author contributions

S.W., and Z.R. developed the concept, designed the experiments and reviewed the manuscript. Y.W. designed and performed the experiments, conducted data analysis, and wrote the manuscript. H.C. performed the experiments and wrote the manuscript. R.S., R.J., and K.L. conducted clinical verification experiments. Y.H. performed the qPCR experiments. Z.G. collected clinical samples. H.W. and L.L. contributed to the experiments. K.M. and D.W. developed the analysis device.

## Competing interests

S.W., Z.R., Y.W., Y.H., H.C. and H.W. are inventors on a pending patent application (no. 202410393528.3) held by the Bioinformatics Center of AMMS. This pending patent covers the single-step RPA-CRISPR/Cas13a assay for monkeypox virus detection and all designed sequences used in this work. All other authors declare no competing interests.
