## [Peer Review File · Nature Communications]

Reviewers' comments:

Reviewer #1 (Remarks to the Author):

Manuscript entitled “Ultrasensitive single-step CRISPR detection of monkeypox virus in minutes with a vest-pocket diagnostic device” reports on a portable Mpox detection system. Experiments are carefully designed, and conclusions are in line with the results. In this study, authors repurposed some previously established systems with significant improvements and innovations. The quantitative nature of the methods and portability are key additions. The manuscript needs improvements and should be revised carefully.

1. Authors failed to cite Mpox detection related recent publications. Contributions in the field should be acknowledged. Simple search shows several recent publications on this topic. Please check papers published in Journal of Medical Virology, Journal of Infection, and Travel Medicine and Infectious Diseases etc.
2. Since Mpox is a DNA virus, the CRISPR-Cas12 system could be used in this study. Why did the authors use Cas13a that needs an additional transcription step? I would appreciate it if authors provide experimental data to justify the use of Cas13a in the current study.
3. Provide more details about the device design, wireless connection, data acquisition, and processing. Are the algorithms developed in this study accessible online? Authors should consider related devices while discussing their device (the key innovative part of the study).
4. There is a cursory description of primer and crRNA selection.
5. Authors claim that virus lysis is better than extraction. Could authors provide more data to support this claim?
6. Include animation(s) to demonstrate device use and assay protocol.
7. Provide more details on pseudovirus preparation.
8. Figure S5: what do authors mean by before and after calibration?
9. Include RPA in scheme 1.
10. Use better quality figures (including supplementary figures).
11. Language should be proofread.

Reviewer #2 (Remarks to the Author):

In this manuscript, the authors present a single-step CRISPR-based assay for detecting the monkeypox virus (MPXV) using a compact diagnostic device called CRISPRPod (CPod). This device can perform viral lysis and a CRISPR-based assay. The authors first optimized the extraction-free viral lysis, reagent lyophilization, and single-step RPA-CRISPR/Cas13a assay, and subsequently integrated the entire detection process into the CPod. Their method was capable of detecting MPXV from mock clinical samples within 15 minutes, with an analytical sensitivity of 0.5 copies/ μL .

The main contribution of this manuscript lies in the optimization of the RPA-CRISPR assay. However, there is limited innovation and scientific discovery that does not warrant publication in Nature Communications. The DNA extraction-free process using chelex-100 has been previously demonstrated by several groups (e.g., RA Lee et al, PNAS 2020, 117(41), p.25722). The assay presented in the current manuscript is largely similar to the PNAS paper, with the primary difference being a more compact instrument, which is not particularly novel. Although the authors demonstrated the performance of their method in the CPod for rapid and sensitive detection of MPXV from mock clinical samples, the main features of this single-step assay and the device require more supporting data and in-depth explanation. Consequently, the reviewer cannot recommend this manuscript for publication in Nature Communications. Additional comments from the reviewer are listed below.

1. Although the speed of the single-step assay is impressive, the reviewer believes more supporting data and in-depth discussion would be helpful to explain how the impressive speed can be accomplished, thereby further elevating the merit of this work. To this end, it would be great if the speed of individual components were first shown. As an example, it would be helpful if the authors show that 50 nM of activated LwaCas13a can cleave 1 μM of fluorescence reporter in the appropriate reaction buffer within a few minutes. It may also be helpful to demonstrate that this RPA can produce sufficient amplicons within a few minutes, perhaps via gel electrophoresis. Such data would strongly support that the whole reaction – including RPA, transcription, Cas13 activation, and trans cleavage of the fluorescence reporter – can indeed plateau within 10 minutes. As for discussion, it would be useful if the authors could elaborate on what they mean by “systematically optimizing the assay conditions to reduce the interference between RPA and CRISPR reactions” and drawing better connections between their results and this mechanism. It would also be helpful if the authors could provide more clarity for screening the primers and crRNAs. For example, Fig. S1 should be illustrated in more detail. The caption says that single-step RPA-CRISPR/Cas13a reaction was performed. For RPA primer screening, which crRNA was used for each primer pair in the single-step RPA-CRISPR/Cas13a reaction? Similarly, which RPA primer was used for crRNA screening? What was the reaction condition for these reactions? Is it the same as an optimized condition from Fig. 2? Finally, more details on the reaction conditions, as well as real-time amplification curves should be provided by Figures 2b – 2g.

2. As CPod represents a new device, it would be helpful to demonstrate its function more thoroughly through supporting videos (in supporting information) and images during its operation (in the main text).

3. The reviewer believes more supporting explanation for the sensitivity of this method are needed. Specifically, earlier works such as miSHERLOCK (reference #36) and POC-CRISPR (doi: 10.1016/j.bios.2021.113390) rely on various concentration techniques for nucleic acid targets to facilitate sensitive detection. On the other hand, this method can achieve better sensitivity while seemingly diluting the sample. Toward providing such an explanation, it would be important to provide

in more detail the original sample volume, the lysis buffer volume, and the sample volume used in the single-step assay. According to the Methods section, mock clinical samples and lysis buffer were mixed in a ratio of 1:9. What was the total lysis reaction volume then? As illustrated in Methods section, was 5 μ L out of entire lysis product used for subsequent CRISPR-based reaction? The reviewer wonders whether only a portion of lysis product was used, which would be the critical factor for downstream assay sensitivity.

4. For the mock clinical samples, the authors must provide information on how they set the threshold for differentiating positive samples with low viral loads from negative samples. This information is critical for supporting 100% concordance between their method and the standard method. For example, in Figure S7, real-time curves from saliva sample #17 and negative samples (#21 - #24) seems to be very similar with their CPod method whereas qPCR assay shows detectable signal increase at \sim 40 cycle with sample #17 but no signal from negative samples. Similarly, signal increase with their CPod method from blood sample #43 also appeared indistinguishable from negative samples (#45 - #48) by eye.

5. The authors compared two sample processing methods, direct lysis and DNA extraction with commercial kit. While more detailed description of direct lysis method needs to be provided, DNA extraction method also needs to be illustrated for systematic comparison between two methods. How much volume of blood and saliva was used for viral DNA extraction, and what was the elution volume? Was 5 μ L out of elution volume used for downstream assay? If so, are dilution factors for downstream CRISPR-based assay identical between two methods?

6. Although the authors stated that there is a quantitative mode in CPod, the quantification capability was mostly an afterthought in the manuscript. The reviewer had a difficult time finding the method for quantification in the manuscript. The reviewer also wonders what the correlation between the target quantity and the quantification method was. The authors should more elaborately discuss the quantitative mode of CPod by addressing the reviewer's questions and providing other relevant information.

7. Although the authors stated that saliva and blood are validated to be effective for MPXV testing, the references provided by the author only confirmed saliva as effective specimen for MPXV detection. On the other hand, according to the WHO, the best diagnostic specimens are swabs directly taken from the rash (skin, fluid, or crusts). Oropharyngeal, anal, or rectal swabs are alternative in the absence of skin lesions, and blood testing is not recommended (<https://www.who.int/news-room/fact-sheets/detail/monkeypox>). While the authors successfully optimized the entire procedure for portable, rapid, and sensitive detection of MPXV with thermal lysis and CRISPR-based assay, it would be more relevant to use real clinical saliva or blood samples from MPXV infected patients unless the authors use mock clinical samples that matches with the recommendation from WHO.

Other minor Comments

1. As far we the reviewer can tell, miSHERLOCK (reference #36) employs a single-step CRISPR-based assay, not a two-step. The authors are asked to double check their classification of miSHERLOCK in Table S2 and make changes if necessary.

2. The authors should improve the readability of several figures in the main text and in the supplementary information, including but not limited to Figures 3e, 4, and S2.

3. For potential reproducibility and dissemination, it would be helpful if the authors could provide more information on the development of their smartphone application (e.g., program code).
4. The manuscript could benefit from another round of edits to correct minor issues such as defining LFA in the introduction and explicitly stating that F3L refers to the target gene.

Reviewer #1:

Manuscript entitled “Ultrasensitive single-step CRISPR detection of monkeypox virus in minutes with a vest-pocket diagnostic device” reports on a portable Mpox detection system. Experiments are carefully designed, and conclusions are in line with the results. In this study, authors repurposed some previously established systems with significant improvements and innovations. The quantitative nature of the methods and portability are key additions. The manuscript needs improvements and should be revised carefully.

Comment 1: Authors failed to cite Mpox detection related recent publications. Contributions in the field should be acknowledged. Simple search shows several recent publications on this topic. Please check papers published in Journal of Medical Virology, Journal of Infection, and Travel Medicine and Infectious Diseases etc.

Response: We feel great thanks for your professional suggestion. We have systematically compared the Mpox detection related recent publications (Table S3). Our developed system showed the lowest LoD and shortest turn-around time. Besides, our assay induced viral lysis and reagent lyophilization, making it easy-to-use, time-saving, and suitable for on-site use. Furthermore, most of the existing studies were lacking of clinical verification, whereas we tested 102 clinical samples (35 Mpox cases, 19 HSV cases, 48 negative cases) with superior clinical feasibility.

Comment 2: Since Mpox is a DNA virus, the CRISPR-Cas12 system could be used in this study. Why did the authors use Cas13a that needs an additional transcription step? I would appreciate it if authors provide experimental data to justify the use of Cas13a in the current study.

Response: Thanks for your professional advice. We further analyzed the reason for the high sensitivity of our assay. Currently, most CRISPR-Dx methods utilized Cas12a for DNA target detection because Cas12a could directly target DNA amplicon without an additional transcription step. Nevertheless, as for single-step RPA-CRISPR reaction, the trans-cleavage activity of Cas12a with the assistance of crRNA could be activated once in the presence of DNA target, whereas the cis-cleavage activity will be also

activated, leading to the unwanted cleavage of DNA target. In the early stage of DNA amplification, the loss of target templates will hinder the nucleic acid amplification and finally result in poor detection efficiency (**Fig. S14a**). As a contrast, Cas13a has been proved with stronger collateral activity. Besides, in the single-step CRISPR reaction, Cas13a and crRNA complex targeted the transcribed RNA, therefore the activated cis-cleavage activity just cut the RNA target, which will not affect DNA templates or amplicons (**Fig. S14b**). Furthermore, physical or chemical isolation of nucleic acid amplification from Cas detection has been validated could significantly improve the reaction sensitivity. The additional T7 transcription step could delay the activation of Cas13a to weaken the interference between RPA and Cas13a cleavage. Thus, we believed Cas13a was preferable to Cas12a in single-step CRISPR reaction for DNA target detection. We hope the above analysis and explanation for the utilization of Cas13 in this study can resolve your concerns.

a

b

Fig. S14. The schemes of detection mechanisms of (a) single-step RPA-CRISPR/Cas12 and single-step RPA-CRISPR/Cas13.

Comment 3: Provide more details about the device design, wireless connection, data acquisition, and processing. Are the algorithms developed in this study accessible online? Authors should consider related devices while discussing their device (the key innovative part of the study).

Response: Thank you for your concern. The circuit diagrams and printed circuit board of CPod were provided in **Fig. S3-S4**. The code of the Android smartphone application is now available at <https://github.com/zayden/cpod>.

Most of the existing detection methods use bulky and expensive benchtop equipment to collect fluorescent signals (Shi *et al.*, *Sci. Adv.* 2021,7: eabc7802; Li L, *et al.*, *ACS Synth Biol* 209, 8, 2228-2237; Casati B, *et al.*, *Nat. Commun.* 2022, 13, 3308; Lu S, *et al.*, *Nat. Biomed. Eng.* 2022, 6, 286-297), which were not suitable for the on-site use. Some portable or handheld homemade devices with filter-based optical designs in recent reports were still costly (Fozouni *et al.*, 2021, *Cell* 184, 1–11; Yan, H. *et al.*, *Nat. Biomed. Eng* 2023, 7, 1583–1601; Ning *et al.*, *Sci. Adv.* 2021; 7 : eabe3703). Besides, devices embedded with smartphone as the imager also require a high cost. Thus, we believe the analysis device with filter-free design and standalone or wireless control mode were urgently needed for fluorescence detection. Here, we developed a miniaturized, low-cost, user-friendly, and multifunctional homemade device named CPod to adapt the above CRISPR-based detection assay. The cost and size of the device were significantly reduced by replacing the optical filter with a spectral sensor (AS7341). Currently, only the 500-520 nm (FAM) optical channel of the AS7341 sensor was utilized for MPXV detection. The advantage of this spectral sensor-based device will be further highlighted for expanded multiplex analysis applications in future. The overall dimension of the homemade CPod was 52 mm × 30 mm × 60 mm. The total cost of the optical, mechanical, and electronic components was 13.46 dollars. Unlike the reported smartphone-embedded devices, users can control the CPod and interpret the results via their own smartphone installed with the application. Besides, CPod can

still work without smartphone. Users can operate CPod just through the buttons and interpret the detection results through indicator lights. Given its excellent characteristics, we believe this miniaturized, simple-to-use, and low-cost device possessed great application prospects in primary medical facilities.

Comment 4: There is a cursory description of primer and crRNA selection.

Response: We feel sorry for our careless description of primer and crRNA selection. We have revised the description of primer and crRNA selection in manuscript, and the agarose gel electrophoresis and Cas13a detection experiments were added to further validate the RPA primers and crRNA screening. The results were shown in **Fig. S1**. In brief, we screened 16 groups of RPA primers, and the agarose gel electrophoresis indicated that the F2-R4 group exhibited the best amplification efficiency (**Fig. S1a**). Then, 3 different crRNA targeting MPXV F3L gene were designed and tested by the transcribed and purified F3L gene RNA. As illustrated in **Fig. S1b**, crRNA 1 demonstrated the optimal cleavage activity. Nevertheless, when we conducted the primers and crRNA screening simultaneously in single-step RPA-CRISPR/Cas13a reaction, we surprisingly found that the group of best primers (F2-R4) and best crRNA (crRNA 1) didn't show the best detection efficiency. And the group of suboptimal primers (F3-R1) and suboptimal crRNA (crRNA 3) worked best (**Fig. S1c**). We guessed it may stem from two aspects. 1) The single-strand DNA of primers and single-strand RNA of crRNA may induce mutual interference; 2) The efficiency of amplification and Cas cleavage should meet the balance. Thus, we recommended the primers and crRNA screening of single-step CRISPR should be conducted simultaneously to obtain more sensitive detection.

Fig. S1. Primers and crRNAs screening. (a) The gel image of 16 pairs of RPA primers screening via 3% agarose gel electrophoresis. (b) Real-time fluorescence curves of Cas13a reaction for 3 crRNA screening. (c) Real-time fluorescence curves of RPA primers and Cas13a crRNA simultaneous screening were tested by plate reader using the single-step RPA-CRISPR/Cas13a reaction for 20 minutes. Target: MPXV F3L gene of 10 copies/ μ L.

Comment 5: Authors claim that virus lysis is better than extraction. Could authors

provide more data to support this claim?

Response: Thanks for your professional suggestion. To further test the performance of viral lysis, we have tested our proposed viral lysis method with real clinical samples, including rash fluid swab, oral swab, saliva, and urine. Furthermore, we systematically optimized the lysis time (**Fig. 3a-d**), lysis temperature (**Fig. 3e-h**), ratio of lysis buffer to sample (**Fig. 3i-m**), and volumes of lysed sample (**Fig. 3n-r**). We finally compared our developed extraction-free viral lysis method with nucleic acids extraction. Compared with the multi-steps and time-consuming (>20 min) operation of nucleic acid extraction, our developed viral lysis only needed a 2-min incubation, demonstrating a superior potential for on-site use (**Fig. 3s**). The results in **Fig. 3t-v** suggested that the direct lysis method obtained the similar result to the conventional nucleic acid extraction method for rash fluid swab, oral swab, and saliva. However, due to the complex composition of urine, the lysis effect was remarkably lower than that of nucleic acid extraction (**Fig. 3w**).

Fig. 3 Optimization of extraction-free viral lysis. The real-time fluorescence curves and their end-point fluorescence intensity of the 10-minutes single-step CRISPR reaction using the lysed (a) rash fluid swab, (b) oral swab, (c) saliva and (d) urine samples with different lysis temperature (room temperature, 60, 80, 95, and 98 °C). Lysis time: 5 minutes. The real-time fluorescence curves and their end-point fluorescence intensity of the 10-minutes single-step CRISPR reaction using the lysed (e) rash fluid swab, (f) oral swab, (g) saliva and (h) urine samples with different lysis time (0, 2, 5, 10, and 15 minutes). Lysis temperature: 80°C. The real-time fluorescence curves of the 10-minutes single-step CRISPR reaction using the lysed (i) rash fluid swab, (j) oral swab, (k) saliva and (l) urine samples and their (m) end-point fluorescence intensity with different ratios

of lysis buffer to sample (10:0, 9:1, 3:1, 1:1, 1:3, 1:9, 0:10). The real-time fluorescence curves of the 10-minutes single-step CRISPR reaction loading with different volumes of lysed (n) rash fluid swab, (o) oral swab, (p) saliva and (q) urine sample (0, 1, 2, 5, 7 μ L), and their (r) end-point fluorescence intensity. (s) meant that the viral lysis was easy-to-use and time-saving than the traditional nucleic acids extractions. The real-time fluorescence curves of the 10-minutes single-step CRISPR reaction using (t) rash fluid swab, (u) oral swab, (v) saliva and (w) urine samples processed by viral lysis and nucleic acid extraction respectively.

Comment 6: Include animation(s) to demonstrate device use and assay protocol.

Response: We sincerely appreciate the valuable comments. To better demonstrate device use and assay protocol, we have provided pictures in **Fig. 1b** and the videos (standalone mode and wireless control mode) in Supplementary Materials to show the operation workflow. We sincerely hope that the video will help you better understand the protocol of our proposed system in this study.

Comment 7: Provide more details on pseudovirus preparation.

Response: Thanks for your concern. All the pseudoviruses were purchased from commercial sources. The MPXV, cowpox, vaccinia, and variola F3L gene pseudovirus used in this study were synthesized by Fubio Biological Technology, Co., Ltd. (Shanghai, China) with quantitative analysis by digital PCR. The sequences of the above four F3L gene pseudovirus were listed in **Table S6**. The MPXV pseudovirus standard reference used for sensitivity and reproducibility determination was purchased from the National Institute of Metrology, China.

Comment 8: Figure S5: what do authors mean by before and after calibration?

Response: Thanks for your attention to the signal calibration. As shown in current **Fig. S6**, due to the deviation from experiment to experiment, the baselines of fluorescence intensity in each experiment were inconsistent which lead to an error in end-point interpretation. Thus, the results were calibrated by the software algorithm to remove the baseline offset.

Comment 9: Include RPA in Scheme 1.

Response: Thanks for your professional advice. We have included RPA and T7 transcription in **Fig. 1** to better demonstrate the single-step RPA-CRISPR/Cas13a reaction.

Fig. 1. Point-of-care testing of MPXV using SCOPE. (a) Schematic illustration of SCOPE for MPXV detection. Rash fluid swab, oral swab, saliva, and urine samples were collected and processed with extraction-free viral lysis on the CPod at 80°C for 2 minutes. The exposed MPXV DNA was conducted with single-step RPA-CRISPR/Cas13a reaction at 37°C for 10 minutes. Qualitative interpretation could be made through the indicator lights of the CPod, or quantitative interpretation was carried out by a smartphone application via wireless connection. (b) The operation workflow of SCOPE for MPXV detection.

Comment 10: Use better quality figures (including supplementary figures).

Response: Thanks for your suggestion. We have provided higher quality pictures in the revised manuscript and supplementary materials to enhance readability.

Comment 11: Language should be proofread.

Response: Thanks for your suggestion. We have tried our best to polish the language in the revised manuscript.

Reviewer #2:

In this manuscript, the authors present a single-step CRISPR-based assay for detecting the monkeypox virus (MPXV) using a compact diagnostic device called CRISPRPod (CPod). This device can perform viral lysis and a CRISPR-based assay. The authors first optimized the extraction-free viral lysis, reagent lyophilization, and single-step RPA-CRISPR/Cas13a assay, and subsequently integrated the entire detection process into the CPod. Their method was capable of detecting MPXV from mock clinical samples within 15 minutes, with an analytical sensitivity of 0.5 copies/ μ L.

The main contribution of this manuscript lies in the optimization of the RPA-CRISPR assay. However, there is limited innovation and scientific discovery that does not warrant publication in Nature Communications. The DNA extraction-free process using chelex-100 has been previously demonstrated by several groups (e.g., RA Lee et al, PNAS 2020, 117(41), p.25722). The assay presented in the current manuscript is largely similar to the PNAS paper, with the primary difference being a more compact instrument, which is not particularly novel. Although the authors demonstrated the performance of their method in the CPod for rapid and sensitive detection of MPXV from mock clinical samples, the main features of this single-step assay and the device require more supporting data and in-depth explanation. Consequently, the reviewer cannot recommend this manuscript for publication in Nature Communications. Additional comments from the reviewer are listed below.

Comment 1: Although the speed of the single-step assay is impressive, the reviewer believes more supporting data and in-depth discussion would be helpful to explain how the impressive speed can be accomplished, thereby further elevating the merit of this work. To this end, it would be great if the speed of individual components were first shown. As an example, it would be helpful if the authors show that 50 nM of activated LwaCas13a can cleave 1 μ M of fluorescence reporter in the appropriate reaction buffer within a few minutes. It may also be helpful to demonstrate that this RPA can produce sufficient amplicons within a few minutes, perhaps via gel electrophoresis. Such data would strongly support that the whole reaction – including RPA, transcription, Cas13

activation, and trans cleavage of the fluorescence reporter – can indeed plateau within 10 minutes. As for discussion, it would be useful if the authors could elaborate on what they mean by “systematically optimizing the assay conditions to reduce the interference between RPA and CRISPR reactions” and drawing better connections between their results and this mechanism. It would also be helpful if the authors could provide more clarity for screening the primers and crRNAs. For example, Fig. S1 should be illustrated in more detail. The caption says that single-step RPA-CRISPR/Cas13a reaction was performed. For RPA primer screening, which crRNA was used for each primer pair in the single-step RPA-CRISPR/Cas13a reaction? Similarly, which RPA primer was used for crRNA screening? What was the reaction condition for these reactions? Is it the same as an optimized condition from Fig. 2? Finally, more details on the reaction conditions, as well as real-time amplification curves should be provided by Figures 2b – 2g.

Response: We sincerely appreciate your professional suggestions. As advised, we have validated RPA can produce sufficient amplicons within 10 minutes via agarose gel electrophoresis (**Fig. S1a**) and the 50 nM of activated LwaCas13a can cleave 1 μ M of fluorescence reporter within a few minutes (**Fig. S1b**). For the single-step CRISPR reaction, the primer and crRNA screening significantly determine the detection sensitivity. First, we screened 16 groups of RPA primers, and the agarose gel electrophoresis indicated that the F2-R4 group exhibited the best amplification efficiency (**Fig. S1a**). Then, 3 different crRNA targeting MPXV F3L gene were designed and tested by the transcribed and purified F3L gene RNA. As illustrated in **Fig. S1b**, crRNA 1 demonstrated the optimal cleavage activity. Nevertheless, when we conducted the primers and crRNA screening simultaneously in single-step RPA-CRISPR/Cas13a reaction, we surprisingly found that the group of best primers (F2-R4) and best crRNA (crRNA 1) didn't show the best detection efficiency. And the group of suboptimal primers (F3-R1) and suboptimal crRNA (crRNA 3) worked best (**Fig. S1c**). We supposed it may stem from two aspects. 1) The single-strand DNA of primers and sing-strand RNA of crRNA may induce mutual interference; 2) The efficiency of amplification and Cas cleavage should meet the balance. Thus, we recommended the

primers and crRNA screening of single-step CRISPR should be conducted simultaneously to obtain more sensitive detection. Furthermore, we have provided the real-time amplification curves of optimization of single-step CRISPR (**Fig. S1c**).

Fig. S1. Primers and crRNAs screening. (a) The gel image of 16 pairs of RPA primers screening via 3% agarose gel electrophoresis. (b) Real-time fluorescence curves of Cas13a reaction for 3 crRNA screening. (c) Real-time fluorescence curves of RPA primers and Cas13a crRNA simultaneous screening were tested by plate reader using the single-step RPA-CRISPR/Cas13a reaction for 20 minutes. Target: MPXV F3L gene

of 10 copies/ μL .

Comment 2: As CPod represents a new device, it would be helpful to demonstrate its function more thoroughly through supporting videos (in supporting information) and images during its operation (in the main text).

Response: Thanks for your professional suggestion. We have provided images of user workflow (Fig. 1b) and the supporting videos (standalone mode and wireless control mode) in supporting information to demonstrate the assay protocol of our proposed assay protocol.

Fig. 1b. The operation workflow of SCOPE for MPXV detection.

Comment 3: The reviewer believes more supporting explanation for the sensitivity of this method are needed. Specifically, earlier works such as miSHERLOCK (reference #36) and POC-CRISPR (doi: 10.1016/j.bios.2021.113390) rely on various concentration techniques for nucleic acid targets to facilitate sensitive detection. On the other hand, this method can achieve better sensitivity while seemingly diluting the sample. Toward providing such an explanation, it would be important to provide in more detail the original sample volume, the lysis buffer volume, and the sample volume used in the single-step assay. According to the Methods section, mock clinical samples and lysis buffer were mixed in a ratio of 1:9. What was the total lysis reaction volume then? As illustrated in Methods section, was 5 μL out of entire lysis product used for subsequent CRISPR-based reaction? The reviewer wonders whether only a portion of lysis product was used, which would be the critical factor for downstream assay sensitivity.

Response: We sincerely appreciate your attention for the excellent detection sensitivity

of our method. We further analyzed the reason for the high sensitivity of our assay. Currently, most CRISPR-Dx methods utilized Cas12a for DNA target detection because Cas12a could directly target DNA amplicon without an additional transcription step. Nevertheless, as for single-step RPA-CRISPR reaction, the trans-cleavage activity of Cas12a with the assistance of crRNA could be activated once in the presence of DNA target, whereas the cis-cleavage activity will be also activated, leading to the unwanted cleavage of DNA target. In the early stage of DNA amplification, the loss of target templates will hinder the nucleic acid amplification and finally result in poor detection efficiency (**Fig. S14a**). As a contrast, Cas13a has been proved with stronger collateral activity. Besides, in the single-step CRISPR reaction, Cas13a and crRNA complex targeted the transcribed RNA, therefore the activated cis-cleavage activity just cut the RNA target, which will not affect DNA templates or amplicons (**Fig. S14b**). Furthermore, physical or chemical isolation of nucleic acid amplification from Cas detection has been validated could significantly improve the reaction sensitivity. The additional T7 transcription step could delay the activation of Cas13a to weaken the interference between RPA and Cas13a cleavage. Thus, we believed Cas13a was preferable to Cas12a in single-step CRISPR reaction for DNA target detection.

Fig. S14. The schemes of detection mechanisms of (a) single-step RPA-CRISPR/Cas12 and single-step RPA-CRISPR/Cas13.

Besides, according to your professional advice, we have further tested and optimized the viral lysis in this study using the Mpxv clinical sample. In brief, we tested a series of ratios of lysis buffer to four types of specimens (rash fluid swab, oral swab, saliva, and urine), including 10:0, 9:1, 3:1, 1:1, 1:3, 1:9, 0:10. We determined the optimal ratio of lysis buffer to rash fluid swab was 9:1, oral swab was 3:1, saliva was 1:3, and urine was 9:1 (**Fig. 3i-m**). Furthermore, to evaluate the performance of our proposed assay with different lysed sample volume, we performed the single-step RPA-CRISPR/Cas13a reaction with 0, 1, 2, 5, and 7 μL of lysed samples. And all the four types of specimens exhibited the best reaction efficiency when using 7 μL of lysed samples (**Fig. 3n-r**). The final total lysis reaction volume was 70 μL .

Fig. 3. The real-time fluorescence curves of the 10-minutes single-step CRISPR reaction using the lysed (i) rash fluid swab, (j) oral swab, (k) saliva and (l) urine samples and their (m) end-point fluorescence intensity with different ratios of lysis buffer to sample (10:0, 9:1, 3:1, 1:1, 1:3, 1:9, 0:10). The real-time fluorescence curves of the 10-minutes single-step CRISPR reaction loading with different volumes of lysed (n) rash fluid swab, (o) oral swab, (p) saliva and (q) urine sample (0, 1, 2, 5, 7 μ L), and their (r) end-point fluorescence intensity.

Comment 4: For the mock clinical samples, the authors must provide information on how they set the threshold for differentiating positive samples with low viral loads from negative samples. This information is critical for supporting 100% concordance between their method and the standard method. For example, in Figure S7, real-time curves from saliva sample #17 and negative samples (#21 - #24) seems to be very similar with their CPod method whereas qPCR assay shows detectable signal increase at ~40 cycle with sample #17 but no signal from negative samples. Similarly, signal increase with their CPod method from blood sample #43 also appeared indistinguishable from negative samples (#45 - #48) by eye.

Response: Thanks for your professional advice. In this study, the threshold value was set to the mean of 20 NTC plus three times the standard deviation as 515. You mentioned the result of mock saliva sample #17 detection was similar to the mock saliva negative samples. In this study, we pursued a shorter reaction time, which made the real-time curve seem similar to the negative. But assisting by the excellent performance

of fluorescence collection of our fabricated CPod, the result can be successfully interpreted through end-point fluorescence intensity.

Comment 5: The authors compared two sample processing methods, direct lysis and DNA extraction with commercial kit. While more detailed description of direct lysis method needs to be provided, DNA extraction method also needs to be illustrated for systematic comparison between two methods. How much volume of blood and saliva was used for viral DNA extraction, and what was the elution volume? Was 5 μL out of elution volume used for downstream assay? If so, are dilution factors for downstream CRISPR-based assay identical between two methods?

Response: Thanks for your professional suggestion. For viral DNA extraction, 1 mL of collected clinical sample was firstly centrifuged at 14000 g for 5 min, then gently removed 800 μL of supernatant. The remaining deposit was extracted by a commercial extraction kit. The released DNA was finally eluted with 200 μL of DNase/RNase water (without dilution). For viral lysis, 7 μL of collected clinical sample was mixed with chelex-100-based lysis buffer in a ratio of 1:9 for rash fluid swab, 1:3 for oral swab, 3:1 for saliva, and 1:9 for urine and incubated at 80 $^{\circ}\text{C}$ for 2 min. Thus, the clinical rash fluid swab, oral swab, saliva, and urine samples were diluted by a factor of 10, 4, 1.33, and 10, respectively. Even though, the fluorescence intensity curves for rash fluid swab, oral swab, and saliva samples processed via viral lysis were similar to that of viral DNA extraction results. While, the result of urine sample processed via viral lysis was weaker than that of viral DNA extraction result. These results validated the excellent nucleic acid release efficiency of the proposed viral lysis method for various clinical sample types.

Fig. 3 (s) meant that the viral lysis was easy-to-use and time-saving than the conventional nucleic acids extractions. The real-time fluorescence curves of the 10-minutes single-step CRISPR reaction using (t) rash fluid swab, (u) oral swab, (v) saliva and (w) urine samples processed by viral lysis and nucleic acid extraction, respectively.

Comment 6: Although the authors stated that there is a quantitative mode in CPod, the quantification capability was mostly an afterthought in the manuscript. The reviewer had a difficult time finding the method for quantification in the manuscript. The reviewer also wonders what the correlation between the target quantity and the quantification method was. The authors should more elaborately discuss the quantitative mode of CPod by addressing the reviewer's questions and providing other relevant information.

Response: We sincerely appreciated your professional suggestion. We have performed a nonlinear regression fitting for the sensitivity testing results of CPod (**Fig. S8**). Thus, we could estimate the input target concentration based on the fitting equation and realize the semi-quantification for the MPXV detection.

Fig. S8. The nonlinear regression fitting for the sensitivity testing results of CPod.

Comment 7: Although the authors stated that saliva and blood are validated to be effective for MPXV testing, the references provided by the author only confirmed saliva as effective specimen for MPXV detection. On the other hand, according to the WHO, the best diagnostic specimens are swabs directly taken from the rash (skin, fluid, or crusts). Oropharyngeal, anal, or rectal swabs are alternative in the absence of skin lesions, and blood testing is not recommended (<https://www.who.int/news-room/fact-sheets/detail/monkeypox>). While the authors successfully optimized the entire procedure for portable, rapid, and sensitive detection of MPXV with thermal lysis and CRISPR-based assay, it would be more relevant to use real clinical saliva or blood samples from MPXV infected patients unless the authors use mock clinical samples that matches with the recommendation from WHO.

Response: We sincerely appreciated your professional suggestion. Considering the

specimen types recommended by WHO, we finally selected rash fluid swab, oral swab, saliva, and urine as detection specimens and collected 102 cases of corresponding real clinical samples. As results shown in **Fig. 6**, our proposed method demonstrated 100% sensitivity and 100% specificity in all four types of specimen and was 100% concordant with the real-time PCR, which demonstrated our proposed with excellent clinical feasibility.

Fig. 6. Validation of the clinical performance of SCOPE. (a) Workflow of qPCR assay and our CRISPR-based assay for testing of mock clinical samples. (b) The corresponding fluorescent intensities at 10 min and its (c) receiver operating characteristic curve for 102 clinical samples tested by SCOPE. (d) Clinical samples RT-qPCR C_t value plotted against fluorescent intensity readout by CPod. (e) Correspondence of qPCR and SCOPE results for the MPXV positive samples, HSV samples, and negative samples.

Other minor comments

Minor Comment 1: As far as the reviewer can tell, miSHERLOCK (reference #36) employs a single-step CRISPR-based assay, not a two-step. The authors are asked to double check their classification of miSHERLOCK in Table S2 and make changes if necessary.

Response: We apologize for this mistake. We have revised it in the **Table. S1**.

Minor Comment 2: The authors should improve the readability of several figures in the main text and in the supplementary information, including but not limited to Figures 3e, 4, and S2.

Response: Thanks for your suggestion. We have provided the higher quality figures in main text and supplementary information.

Minor Comment 3: For potential reproducibility and dissemination, it would be helpful if the authors could provide more information on the development of their smartphone application (e.g., program code).

Response: Thanks for your suggestion. We have provided the smartphone application interface in **Fig. S7**. The code of the Android smartphone application is now available at <https://github.com/zayden/cpod>.

Fig. S7. The smartphone software interface for the wireless control of CPod.

Minor Comment 4: The manuscript could benefit from another round of edits to correct minor issues such as defining LFA in the introduction and explicitly stating that F3L refers to the target gene.

Response: Thanks for your suggestion. We have carefully checked and corrected the description in the manuscript.

REVIEWERS' COMMENTS

Reviewer #1 (Remarks to the Author):

Authors have carefully addressed all my major concerns/comments. The revised version is much improved now. Authors have added clinical data that further adds value to manuscript quality.

Reviewer #2 (Remarks to the Author):

In this revised manuscript, the authors have enhanced the presentation of their SCOPE for rapid and sensitive MPXV detection. The reviewer appreciates the authors' efforts in addressing the reviewer's comments and improving their manuscript. Nevertheless, the reviewer still has several comments and concerns that the reviewer would like to highlight regarding this work, as described below.

1. While the authors claimed the sensitivity of SCOPE as 0.5 copies/ μ L, it is the sensitivity of the single-step RPA-CRISPR/Cas13a reaction without accounting for the lysis step. Given that the lysis step involves sample dilution with varying dilution factors, the sensitivity of SCOPE would be higher depending on the dilution factor.
2. Although the authors answered the reviewer's major comment #4 and provided the information of how they set the threshold, it was still not provided in the revised manuscript. It would be helpful if the authors incorporate this information into the manuscript.
3. As the lysis protocol was shown to be a critical factor to achieve rapid and sensitive MPXV detection in SCOPE, it would be important to demonstrate how 7 μ L of directly lysed samples showed similar fluorescence increases to the extracted viral DNAs from 1 mL samples (Fig 3t-v). Assuming 100% lysis efficiency in the direct lysis, the yield of commercial extraction kit needs to be only 0.7% in order to obtain similar fluorescence increases from these two methods. The reviewer wonders about the validity of considering a 0.7% extraction yield as a representative estimation for commercial extraction kit.

Responses to Reviewers

Reviewer #1 (Remarks to the Author):

Authors have carefully addressed all my major concerns/comments. The revised version is much improved now. Authors have added clinical data that further adds value to manuscript quality.

Response: Thank you for your comment. We appreciate your time and attention.

Reviewer #2 (Remarks to the Author):

In this revised manuscript, the authors have enhanced the presentation of their SCOPE for rapid and sensitive MPXV detection. The reviewer appreciates the authors' efforts in addressing the reviewer's comments and improving their manuscript. Nevertheless, the reviewer still has several comments and concerns that the reviewer would like to highlight regarding this work, as described below.

Comment 1: While the authors claimed the sensitivity of SCOPE as 0.5 copies/ μL , it is the sensitivity of the single-step RPA-CRISPR/Cas13a reaction without accounting for the lysis step. Given that the lysis step involves sample dilution with varying dilution factors, the sensitivity of SCOPE would be higher depending on the dilution factor.

Response: We sincerely appreciated your professional comment. In this study, we firstly tested the sensitivity of single-step RPA-CRISPR/Cas13a by using different concentration of monkeypox virus DNA extracted by a commercial kit. We quite agreed with your opinion that the sensitivity of SCOPE would be affected by the dilution factor during the lysis step. However, digital PCR instruments are inaccessible in the authorized test laboratory to exactly quantify monkeypox in clinical samples. The semiquantitative results of qPCR demonstrated that SCOPE can detect mock saliva samples with Ct values up to 41.86 (**Fig. S10**), and clinical samples with Ct values ranged from 15.88 to 39.02 (**Fig. 6**). Thus, we believe the proposed SCOPE platform can achieve similar sensitivity to the standard qPCR assay for practical clinical applications.

Comment 2: Although the authors answered the reviewer's major comment #4 and provided the information of how they set the threshold, it was still not provided in the revised manuscript. It would be helpful if the authors incorporate this information into the manuscript.

Response: We sincerely appreciated your professional suggestion. We have revised and provided the information of how we set the threshold in the revised manuscript.

Comment 3: As the lysis protocol was shown to be a critical factor to achieve rapid and sensitive MPXV detection in SCOPE, it would be important to demonstrate how 7 μL of directly lysed samples showed similar fluorescence increases to the extracted viral DNAs from 1 mL samples (Fig 3t-v). Assuming 100% lysis efficiency in the direct lysis, the yield of commercial extraction kit needs to be only 0.7% in order to obtain similar fluorescence increases from these two methods. The reviewer wonders about the validity of considering a 0.7% extraction yield as a representative estimation for commercial extraction kit.

Response: Thanks for your attention on the viral lysis performance. We speculated that there may be the following reasons for the performance of viral lysis similar to the commercial extraction: First, although 1 mL samples were centrifuged and removed 800 μL of supernatant, the effort of this enrichment method was poor. The remaining 200 μL of samples were finally eluted with 200 μL DNase/RNase-free water without enrichment. Besides, some target nucleic acids would be lost in the extraction process. Although the viral lysis method diluted the collected samples, it could also

eliminate the sample loss. Second, we guessed the components of our lysis buffer (20% chelex-100, 50 mM TCEP in TE buffer (10mM Tris, 1 mM EDTA (pH 8.0)) may enhance the single-step RPA-CRISPR/Cas13a reaction. Some previous studies have proved that reducing agents like DTT could enhance the activity of CRISPR/Cas reaction (Sci. Adv., 2021, 7:eabh2944; Nat. Commun., 2021, 12:1739). We speculated TCEP with similar chemical properties to DTT in our lysis buffer may also enhance the CRISPR reaction. Our future work will focus on this issue to further improve the assay performance.